# Effect of Addition of Cross-Linked Starch on the Properties of Degraded PBAT Poly(butylene adipate-co-terephthalate) Films

**DOI:** 10.3390/polym15143106

**Published:** 2023-07-21

**Authors:** Denise Agostina Grimaut, Jania Betania Alves da Silva, Paulo Vitor França Lemos, Paulo Romano Cruz Correia, Jamille Santos Santana, Luiggi Cavalcanti Pessôa, Santiago Estevez-Areco, Lucía Mercedes Famá, Silvia Nair Goyanes, Henrique Rodrigues Marcelino, Denilson de Jesus Assis, Carolina Oliveira de Souza

**Affiliations:** 1Graduate Program in Food Science, College of Pharmacy, Federal University of Bahia, Salvador 40170-115, BA, Brazil; denise.grimaut@gmail.com; 2Center for Exact and Technological Sciences, Collegiate of Mechanical Engineering, Federal University of Recôncavo da Bahia, Cruz das Almas 44380-000, BA, Brazil; 3Graduate Program in Chemical Engineering, Polytechnic School, Federal University of Bahia, Salvador 40210-630, BA, Brazil; milepct@hotmail.com (J.S.S.); luiggi.pessoa@ufba.br (L.C.P.); denilsonengal@gmail.com (D.d.J.A.); 4Graduate Program in Biotechnology-Northeast Biotechnology, Federal University of Bahia, Salvador 40110-100, BA, Brazil; oluap_somel@hotmail.com (P.V.F.L.); paulo.romano85@hotmail.com (P.R.C.C.); 5Department of Physics, Laboratory of Polymers and Composite Materials, Faculty of Exact and Natural Sciences, Buenos Aires University, University City, Buenos Aires 1428, Argentina; santi.ea90@gmail.com (S.E.-A.); lfama@df.uba.ar (L.M.F.); goyanes@df.uba.ar (S.N.G.); 6Department of Medicine, College of Pharmacy, Federal University of Bahia, Salvador 40170-115, BA, Brazil; henrique.marcelino@ufba.br; 7School of Exact and Technological Sciences, Salvador University, Salvador 41820-021, BA, Brazil; 8Department of Bromatological Analysis, College of Pharmacy, Federal University of Bahia, Salvador 40170-115, BA, Brazil

**Keywords:** blends, crosslinking, potato starch, ploy(butylene adipate-co-terephthalate)

## Abstract

This work aimed to evaluate the properties of butylene adipate-co-terephthalate (PBAT) degraded after 1800 days of storage (DPBAT) by preparing blends (films) with crosslinked starch (Cm) through extrusion and thermocompression. Different ratios of DPBAT:Cm (70:30, 60:40, and 50:50 m/m) were prepared. The incorporation of Cm into DPBAT significantly changed the properties of the films by making them stiffer (increasing Young’s modulus by up to 50%) and increasing the thermal resistance of DPBAT. The presence of crosslinked starch in the films made them less hydrophobic (with decreased contact angle and increased moisture content), but these parameters did not vary linearly with changes in the content of crosslinked starch in the blend (DPBAT:Cm). The microscopic images show an inhomogeneous distribution of Cm granules in the DPBAT matrix. Thus, the films prepared with PBAT show a significant decrease in their mechanical parameters and heat resistance after long-term storage. However, the preparation of blends of degraded DPBAT with crosslinked starch promoted changes in the properties of the films prepared by thermocompression, which could be useful for disposable packaging.

## 1. Introduction

Increasing concern for environmental protection has stimulated interest in the study of environmentally friendly materials. As a result, concepts such as biodegradability and environmental safety have become trending topics in the field of new material development [1,2,3]. In the last two decades, the industry has prioritized using environmentally safe materials in packaging materials to comply with the increasing number of governmental policies that aim to reduce the amount of waste generated by non-biodegradable plastics. Today, there is a need to develop environmentally friendly and biodegradable materials. Those based on a blend of starch and biodegradable polyesters are the most economically competitive among the various biodegradable plastics available. Starch is a biopolymer derived from renewable resources such as wheat, potatoes, and corn [4,5,6].

Starch-based blends are characterized by their water solubility, hygroscopic behavior, low melting point, faster aging due to retrogradation, and poorer mechanical properties compared to synthetic polymer-based materials [7,8]. Different strategies are used to overcome these weaknesses, such as the mixing of starch with other polymeric materials [9,10,11], the addition of fillers [12,13], or the introduction of chemical substituents such as phosphate, aldehyde, acetate, and carboxyl into the starch molecule. The incorporation of these groups leads to an improvement in mechanical and technological properties [14].

Modifying starch by phosphorylation of the reactive hydroxyl groups on the anhydroglucose units leads to crosslinking, which causes strengthening of the hydrogen bonds between the starch molecules. This makes these starches more resistant to high temperatures, low pH, and high shear forces. Phosphorylation can be achieved through starch reactions with phosphoric acid or aqueous solutions of ortho-, pyro-, or tripolyphosphoric acid salts at controlled pH and temperature. The improvements in native and phosphorylated starch properties have already been investigated [15,16]. In addition, many studies address the potential benefits of blending starch with biodegradable polyesters such as poly(butylene adipate-co-terephthalate) (PBAT) [11,12,17,18,19,20], poly(butylene succinate) (PBS) [21,22], polycaprolactone (PCL) [23,24,25,26], and polylactic acid (PLA) [27,28].

In blends of starch with other polymeric materials, poly(butylene adipate-co-terephthalate) (PBAT) is one of these promising materials because it is a synthetic, fully biodegradable, aliphatic–aromatic co-polyester that combines biodegradability with other desirable physical properties for disposable packaging. The combination of starch and PBAT offsets the disadvantages of plasticized starch by improving mechanical properties and dimensional stability and reducing the hydrophilic nature of starch [20,21,22,23,24,25,26,27,28,29,30]. 

The COVID-19 pandemic, which began at the end of 2019, has had a profound impact on various sectors of society around the world. The removal of researchers from their research centers brought significant challenges, among them a loss of quality in the materials and inputs used for research. Over time, PBAT can undergo degradation, which is defined as a process that describes the cleavage of the polymeric chain into oligomers and, eventually, monomers. This process leads to the loss of mechanical and thermal properties [31]. Therefore, it is urgent to develop alternatives to enable the reuse of this high added value polymer.

There are no studies on obtaining and evaluating blends of degraded PBAT with starch chemically modified by phosphorylation (crosslinking). Therefore, this study aims to prepare film blends through extrusion and subsequent thermocompression of degraded PBAT and starch chemically modified by phosphorylation to improve the properties of the films obtained.

## 2. Materials and Methods

### 2.1. Materials

PBAT polymer was purchased from BASF SE (Ludwigshafen, Germany) under the brand name EcoflexTM. The storage time was 1800 days until use, and it was stored in packaging specified by the manufacturer and protected from light and external humidity. The PBAT after 1800 days of storage was dominated by degraded PBAT (DPBAT) because, according to Guocheng [32], PBAT tends to age in the process of storage, which thus affects different properties, mainly mechanical properties where the tensile strength and the elongation at break show a downward trend, showing that the long storage periods for PBAT (240 days of aging) cause degradation to the internal structure.

Commercial samples of potato starch (*Solanum tuberosum* L.) were acquired from Yoki, Brazil. The potato starch granules presented B-type crystallinity and 20.46% amylose content, as previously characterized by Lemos et al. [33].

All reagents, such as sodium trimetaphosphate (STMP), sodium tripolyphosphate (STPP), sodium sulfate, sodium hydroxide, chloric acid, and sodium phosphate monobasic, were of analytical quality. 

#### Starch Chemical Modification

Potato starch was crosslinked with STMP/STPP (sodium trimetaphosphate and sodium tripolyphosphate, respectively) according to the method described by Woo and Seib [34] and adapted by Lemos et al. [33]. Approximately 100 g of potato starch was dispersed in distilled water (250 g mL^−1^), followed by the addition of sodium sulfate (10% *w*/*w*, based on the dry weight of the starch) and 4.0 g of the crosslinking agents STMP and STPP (99:1). After homogenization, the pH of the mixture was adjusted to 10.5 by adding 1 M sodium hydroxide. The system was homogenized for 1 h at 45 °C in a shaker (180 rpm), and the pH was further adjusted to 5.5 by adding 1 M hydrochloric acid solution. The crosslinked starch was centrifuged (3450 *g*; 15 min) and washed seven times to remove the free phosphorus content. The precipitate was dried in an oven (40 °C) for 24 h and stored.

### 2.2. Film Elaboration 

DPBAT (moisture: 0.39%) and chemically modified starch (Cm) (moisture: 11%) blends were prepared to achieve better dispersion of Cm in the DPBAT matrix (Figure 1). The amounts of the components in the blends were defined based on previous studies [35,36]. 

First, the chemically modified starch was mixed with glycerol (99.5%) at the concentrations shown in Table 1, resulting in a Cm dispersion. To obtain this dispersion, chemically modified starch powder was manually mixed with glycerol until it formed a uniform mixture. In the second step, the dispersion was manually mixed with DPBAT granules. Finally, the different formulations were extruded using an AX plastic twin-screw extruder (model DR1640, AX Plastic, Diadema, SP, Brazil) to obtain pellets. The filament speed was 60 rpm. The temperature program of the zones from the feed (zone 1) to the die zone (zone 8) was 80, 90, 100, 105, 105, 110, 110, and 115 °C. These temperatures were chosen to obtain a completely gelatinized material [36,37].

Later, after extrusion, the thermoplastic pellets were used to produce films through thermocompression with a hydraulic thermopress (Hidraúllicos Moran-15 Tns.). The pellets (~4 g) were placed between Teflon plates and heated to ~130 °C for 15 min. The system’s temperature was then set to 40 °C using the temperature controller. Upon reaching this temperature, the pressure between the presses was increased to 56 kPa. The films based only on DPBAT were prepared following the same conditions and preparatory steps as the formulations of DPBAT:Cm (extrusion and thermocompression of the granules). The resulting films were stored at a relative humidity of 56% for 24 h before being tested according to the method of Morales, Candal, Famá, Goyanes, and Rubiolo [38]. 

### 2.3. Film Characterization 

#### 2.3.1. Morphological Characterization

The morphology of the cryogenic fracture surface of the different film blends was studied through scanning electron microscopy (SEM) using a Zeiss DSM982 Gemini field emission gun (Oberkochen, Germany) at 3 kV. The samples were frozen and fractured under liquid nitrogen, placed on a support, and then sprayed with a thin layer of platinum before observation. 

#### 2.3.2. Moisture Content (MC) and Contact Angle (CA)

Moisture content (MC) was determined using the standard method of AOAC [39], which was also used by Estevez-Areco et al. [40] as described in Equation (1). Samples of each formulation (mw~0.5 g) were dried in an oven at 105 °C for 24 h and then weighed (md). Experiments were performed in triplicate.
(1)$$MC\left(\%\right)=\frac{mw-md}{mw}\times 100$$

An Attention Theta Optical Tensiometer (Biolin Scientific, Gothenburg, Sweden) was used to measure the contact angle (CA) indicated by a small drop of liquid water (~2 μL) resting on a horizontal sample film surface ~1 cm^2^ in size. This method is based on image processing and curve fitting for contact angle measurements, where the angle between the baseline of the droplet and the tangent at the droplet boundary is measured [41]. The external image of the droplet was captured using microlenses and a camera to produce a magnified image of the droplet, allowing for the quantification of changes in droplet shape, which were recorded as digital images over time. The contact angle values in the image were calculated through digital image processing using OneAttension 3.0 software.

#### 2.3.3. Fourier Transform Infrared Spectroscopy (FTIR)

FTIR analyses were performed using a Jasco FT-IR 4100 spectrometer (Jasco International, Tokyo, Japan) equipped with an attenuated total reflectance module (ATR, ZnSe crystal, (Jasco International, Tokyo, Japan)). Spectra were recorded in the range of 4000 to 600 cm^−1^ as an average of 64 scans with a resolution of 4 cm^−1^. 

#### 2.3.4. Thickness and Mechanical Properties 

Films were cut into pieces with dimensions of 35 mm × 5 mm (length and width, respectively). Thickness was measured with a Mitutoyo digital micrometer (0.001 mm resolution). Six random measurements were made on the films, and the thickness was determined arithmetically as an average [35]. Uniaxial tensile tests were performed using a Brookfield Texture Analyzer (CT3-100, Pittsburgh, PA, USA) in accordance with ASTM D882-10 [42]. They were tested at a strain rate of 10^−3^ s^−1^. Representative curves for each system were obtained, and the Young’s modulus (E), stress at break (σ), and strain at break (ε) values were calculated as the average of at least 10 measurements.

#### 2.3.5. Thermogravimetric Analysis (TGA)

Subsequently, 5 mg samples of each film blend were placed in aluminum trays in the TGA balance (Shimadzu, Tokyo, Japan). Tests were performed under a nitrogen atmosphere (flow rate of 30 mL min^−1^) from 30 to 450 °C at a heating rate of 10 °C min^−1^ [36].

### 2.4. Data Processing and Statistical Analysis

The normality of the data was evaluated using the Shapiro–Wilk test, and a comparison of means was performed using analysis of variance (ANOVA) and Tukey’s test. Statistical analyses were performed using Statistica 7.0 software at a 5% significance level.

## 3. Results and Discussion

### 3.1. Morphological Characterization

Morphology is a characteristic directly related to the mechanical and rheological properties of polymeric blends [43]. These characteristics can be justified from observations of the distribution of the dispersed phase (Cm) in the continuous phase (DPBAT). Figure 2 shows cross sections of the film samples from the different images taken during the SEM observations.

Figure 2A shows the surface of DPBAT, which is similar to that obtained by Silva et al. [36] for PBAT before long-term storage. It was also confirmed that DPBAT acted as the continuous phase in all film samples, while Cm was the dispersed phase. Those starch granules that were dispersed in the DPBAT matrix can be observed.

Images of the fracture surface of the films show the presence of two immiscible phases in the mixture (Figure 2B–E); there are granules of Cm that are not completely broken and distributed in the DPBAT matrix, resulting in an inhomogeneous structure. The presence of gaps at the interface between the Cm granules and the DPBAT matrix may indicate a lack of adhesion between the mixture components, even when glycerol is used as a plasticizer. The crosslinking process may increase the resistance and compactness of the starch granules, requiring greater thermomechanical energy (extrusion process) to break them and promote mixing with DPBAT.

Another important factor in the preparation of polymer blends is the ratio between the viscosities of the components (DPBAT/Cm). When the ratio between the viscosity of the dispersed phase and the continuous phase approaches 1, the dispersion of the dispersed phase in the matrix is favored, contributing to a homogeneous blend. In our case, Cm and DPBAT have different viscosities (the viscosity of Cm is probably higher than that of DPBAT because it resembles unmodified starch) and the images are consistent with the theory, although glycerol was added to the formulation to improve the interactions between Cm and DPBAT [36].

### 3.2. Contact Angle (CA)

Water CA can be used to measure the wettability of film mixtures. It is important to emphasize that a low CA indicates greater surface hydrophilicity [41]. To evaluate this property, the contact angle (*θ*) between a drop of distilled water and the surface of each film was tested (Figure 3, Table 2).

The incorporation of Cm and glycerol into DPBAT resulted in a decrease in the CA of the films. This suggests an increase in the hydrophilic character of these blends. PBAT is a polymer with hydrophobic characteristics, whereas starch, even if crosslinked, is a hydrophilic material. Thus, the addition of Cm to PBAT leads to an increase in the sweetness of the blend. However, the decrease in CA was not proportional to the Cm content in the mixture, as shown in Figure 3. The CA values for DPBAT, DPBAT:Cm I, DPBAT:Cm II, and DPBAT:Cm III were 102.39°, 75.25°, 64.59°, and 83.75°, respectively. The discrepancy between the DPBAT:Cm III value and the other values could be due to the selected portion used as the body of proof for analysis, which may have had lower Cm and glycerol content. Although the Cm and glycerol concentrations were the highest, this was not uniform, as shown by microscopy. For all other samples, the contact angle decreased with increasing Cm and glycerol content, confirming the hydrophilic behavior of Cm [12]. This result is consistent with the moisture content data in Table 2.

### 3.3. Fourier Transform Infrared Spectroscopy (FTIR)

As is shown in Figure 4, DPBAT presented spectral bands at 2958–2875 cm^−1^, representing the asymmetric stretch of CH_2_ associated with the aromatic or aliphatic parts [44], at 1712 cm^−1^, representing the stretching of carbonyl groups (C=O) on the ester bonds, at 1578–1504 cm^−1^, representing the skeleton vibration of benzene rings, and at 1456 cm^−1^, which can be attributed to the vibration of C–H in CH_3_ groups. Further, 1169 cm^−1^ and 1121 cm^−1^ refer to the stretching mode of the C–O in carboxylic acid attached to the aromatic ring acid and the ester linkage between the aliphatic parts of the molecule. The band at 731 cm^−1^ represents the bending vibration of the CH plane of the benzene ring. According to Cai et al. [45], bands at 731, 1121, and 1169 cm^−1^ can be used as indicative bands to identify PBAT. 

The DPBAT:Cm (I, II, and III) films (blends) also showed characteristic PBAT bands. In addition, they presented other bands related to Cm. The bands around 2900-3300 cm^−1^ represent water absorption, which can be attributed to the partially hydrophilic character of crosslinked starch [34]. As the spectra show a difference in the intensity of these bands, greater intensity is observed in the film containing the higher concentration of Cm [DPBAT:Cm III]. Additionally, the bands at 1645 cm^−1^ are due to the water present in the Cm molecule [46]. However, intense bands are observed in the films (blends) at 3400 cm^−1^, which is associated with hydrogen bonds between the phosphate groups and the hydroxyl groups in starch and glycerol. Intense bands between 1020 and 880 cm^−1^ referring to O–P–O bonds can be attributed to the starch crosslinking reaction (Cm) [46]. Moreover, there is a broad band in the region of 727 cm^−1^ due to the stretching of CH_2_ groups, as observed by Brandelero et al. [18].

### 3.4. Thickness and Mechanical Properties of Film Blends

The thickness of the films varied depending on the composition of the mixtures. The range of variation was between 133 and 185 μm (Table 2). From these results, it can be suggested that the addition of starch decreased the thickness of the films (blends), although the variation was not linear. However, these results corroborate the mechanical tests performed. Figure 5 shows the stress–strain curves for DPBAT, DPBAT:Cm I, DPBAT:Cm II, and DPBAT:Cm III. The results for maximum stress, Young’s modulus, and elongation at break are shown in Table 2.

The DPBAT showed lower mechanical parameters tensile strength (σ  = 5.47 MPa), elongation at break (ε = 31.94%), and modulus (E = 32.83 MPa), Table 2) than the film of the PBAT without long-term storage (σ = 20 MPa, ε = 468% and E = 93 MPa), specifically a reduction in tensile strength of about 65%, a reduction in elongation at the break of 93%, and a reduction in Young’s modulus of 65% [36]. Other authors reported similar values (σ = 32–36 MPa, ε = 600%, and E = 20–60 MPa) [47,48], demonstrating the degradation of PBAT after a long storage period (1800 days). In a similar study, Guocheng [32] reported the effects of aging time (240 days) on the tensile strength and elongation at the break values of PBAT films and showed a tendency for these parameters to decrease with a reduction in σ of up to 39%. This behavior is attributed to the presence of carbonyl groups in the chemical structure of the polymer, which is oxidized and generates free radicals that accelerate the degradation of the polymer, consequently leading to a decrease in properties.

Adding Cm to DPBAT reduced the σ and ε values of the films compared to DPBAT (σ = 5.47 MPa and ε = 31.94%, respectively). DPBAT:Cm I showed the greatest reduction in these parameters (σ = 1.86 MPa and ε = 7.07%). However, the presence of Cm and glycerol (concentrations of 30 and 40%) in the blend made the films stiffer and led to an increase in Young’s modulus compared to the films prepared with only DPBAT (E = 32.83 MPa). The DPBAT:Cm I film showed an E value of 30.85 MPa, and the DPBAT:Cm II film showed an E value of 39.95 MPa.

The blends of Cm and glycerol in DPBAT changed the mechanical parameters of the films (σ, ε, and E). This behavior can be attributed mainly to the type of starch modification used, as esterification (crosslinking) changes the polymer’s molecular structure, which can lead to increased viscosity and shear strength [32]. The increase in viscosity and shear strength may have contributed to the inhomogeneous dispersion of Cm in DPBAT during extrusion processing (as seen in the microscopic images of the films shown in Section 3.1), resulting in a decrease in the tensile strength and elongation of the films. However, the strength obtained by crosslinking (preferably covalent bonds) favors mechanical strength, which might have contributed to the increased Young’s modulus of the films. This is because the crosslinking process promotes the densification of the structures and restricts the movement of the molecular chains [44,45,46,47,48,49].

### 3.5. Thermogravimetric Analyses (TGAs)

The thermogravimetric analysis curves of DPBAT film and films (blends) are shown in Figure 6. The DPBAT exhibited a single degradation event with an initial temperature of T_onset_ = 332 °C, which can be attributed to polymer degradation (Figure 6; Table 3). Other authors have reported a PBAT degradation temperature of about T_onset_ = 356 °C [35,50,51,52]. The temperature at the onset of thermal degradation of DPBAT decreased by about 24 °C compared to the values reported in the literature, indicating a loss of thermal resistance for this polymer after long storage (1800 days).

The films had two main mass loss events (T1 and T2), with a mass loss of about 90% and low moisture content (0.4–2.49%) (Table 3; Figure 6a). Event T1 was associated with the decomposition of glycerol and derivatized starch, with initial decomposition temperatures (T_onset_) of 267 °C (−16%), 255°C (−33%), and 262 °C (−24%) for DPBAT:Cm films I, II, and III, respectively. The thermal decomposition of starch polymers involves the cleavage of polyhydroxy groups, depolymerization of chains, and the elimination of aldehydes and ketones [44,52]. Starch phosphorylation results in crosslinking between chains through a covalent bond that increases this biopolymer’s thermal stability [15,16,53] when compared to its native counterparts. The second event (T2) is related to the thermal decomposition of the polymeric matrix with DPBAT as the major component. Despite no expressive changes in T1 in the studied samples (Table 3; Figure 6b), the thermal stabilities of the major component were affected by the addition of derivatized starch granules. The films showed a higher onset of degradation (T_onset_ = 347 °C (−77%), 353°C (−62%), and 340 °C (−68%)), indicating that adding Cm to the mixture increased the T_onset_ of DPBAT by up to 20 °C.

## 4. Conclusions

Films prepared with DPBAT showed a significant decrease in mechanical parameters and heat resistance after long-term storage (1800 days) compared to non-degraded PBAT films reported in the literature. The preparation of blends of degraded PBAT with Cm and glycerol promoted changes in the properties of films prepared through thermocompression. The presence of Cm and glycerol in the films made them less hydrophobic, decreased the contact angle, and increased moisture content. However, these parameters did not change proportionally to the modified starch content in the blend (DPBAT:Cm) due to the inhomogeneous distribution of the mixture components, as shown in the microscopic images. Nevertheless, the addition of Cm and glycerol to DPBAT resulted in an increase in the thermal stability and Young’s modulus of the films. The results suggest that it is possible to use degraded PBAT (after long-term storage) in combination with polymers such as crosslinked starch. In terms of prospective studies, the microscopic images show that a more detailed study of the processing conditions (extrusion) is needed to allow for better adhesion/interaction between the crosslinked starch and the degraded PBAT, which can lead to a significant improvement in the analyzed parameters. This study thus contributes towards advances in the frontier of knowledge regarding disposable packaging.

## Figures and Tables

**Figure 1 polymers-15-03106-f001:**
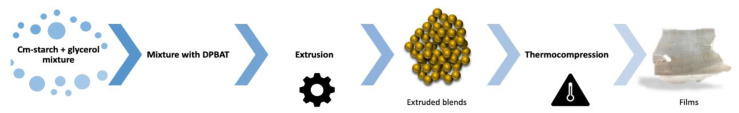
Steps for the preparation of a mixture of poly(butylene adipate-co-terephthalate)/chemically modified starch (DPBAT:Cm) and obtaining films.

**Figure 2 polymers-15-03106-f002:**
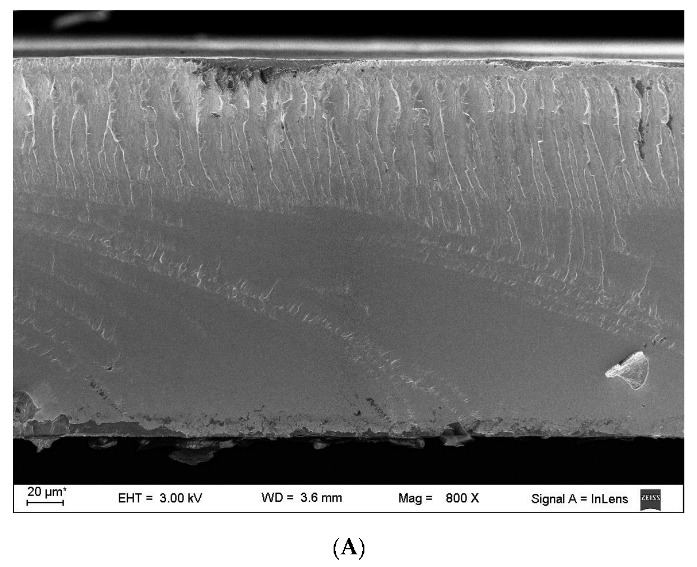
Cross section micrographs of the films: (**A**) DPBAT, (**B**) DPBAT:Cm I, (**C**) DPBAT:Cm II, and (**D**,**E**) surface micrographs of DPABAT:Cm III.

**Figure 3 polymers-15-03106-f003:**
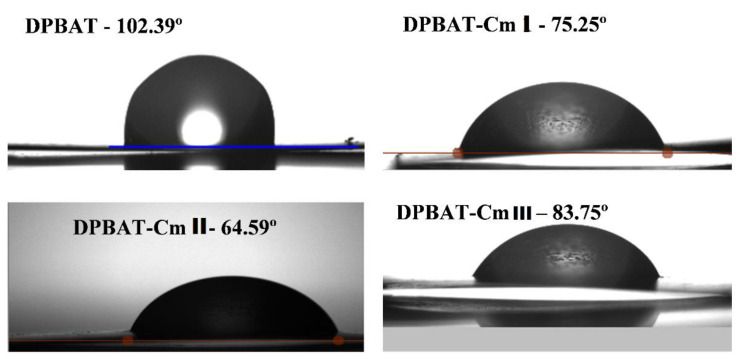
Contact angle (CA) of the film samples.

**Figure 4 polymers-15-03106-f004:**
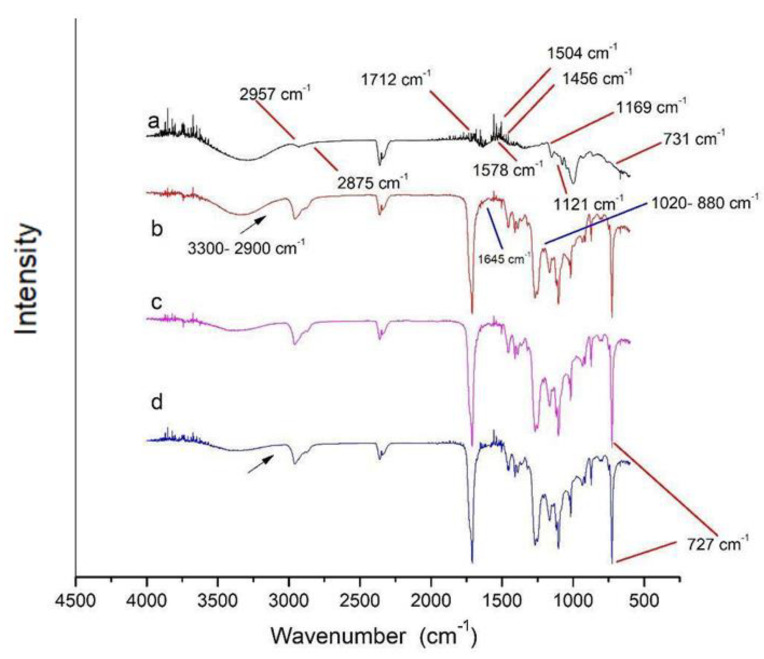
Fourier transform infrared spectroscopy (FTIR) of film samples: (a) DPBAT, (b) DPBAT:Cm III, (c) DPBAT:Cm II, and (d) DPBAT:Cm I.

**Figure 5 polymers-15-03106-f005:**
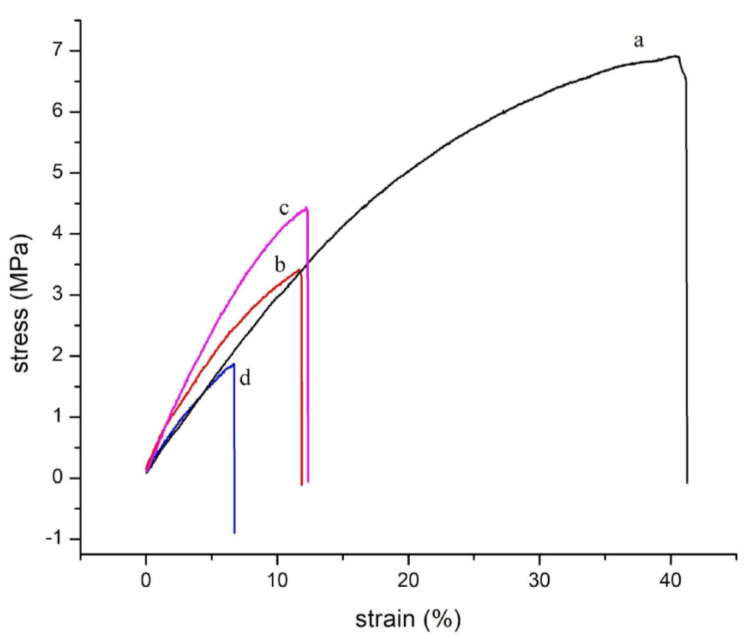
Stress–strain curves for (a) DPBAT, (b) DPBAT:Cm III, (c) DPBAT:Cm II, and (d) DPBAT:Cm I.

**Figure 6 polymers-15-03106-f006:**
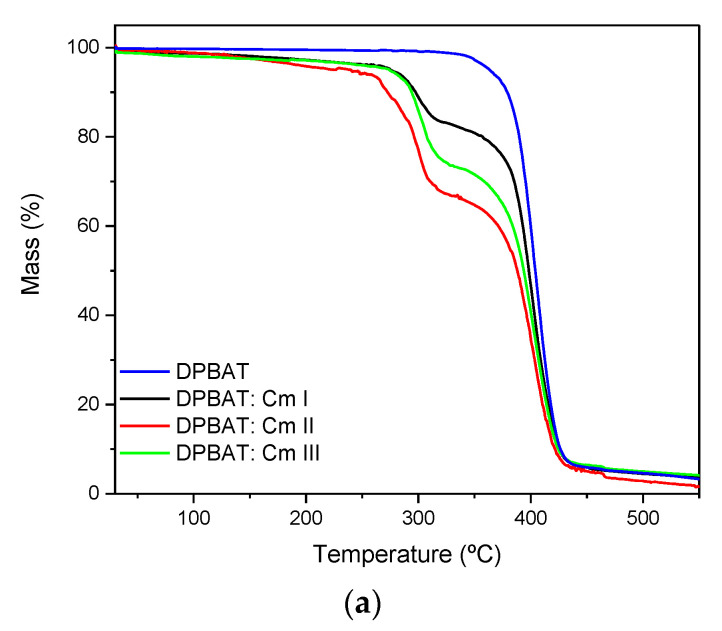
Thermal analysis. (**a**) TG and (**b**) DTG profiles of DPBAT, DPBAT:Cm I, DPBAT:Cm II, and DPBAT:Cm III.

**Table 1 polymers-15-03106-t001:** Composition of blends.

Formulations	Cm (g)	Glycerol (g)	DPBAT (g)
DPBAT	^-^	-	100
DPBAT:Cm I	22.5	7.5	70
DPBAT:Cm II	30.0	10.0	60
DPBAT:Cm III	37.5	12.5	50

Cm, chemically modified starch; DPBAT, degraded PBAT.

**Table 2 polymers-15-03106-t002:** Moisture, contact angle, thickness, tensile strength, elongation at break, and Young’s modulus.

Formulations	Moisture (%)	Contact Angle (°)	Thickness (μm)	Tensile Strength (MPa)	Elongation at Break (%)	Young’s Modulus (MPa)
DPBAT	0.5 ± 0.01 ^a^	102.39	185 ± 45 ^b^	5.47 ± 1.31 ^c^	31.94 ± 9.21 ^a^	32.83 ± 3.94 ^a^
DPBAT:Cm I	4.0 ± 0.04 ^c^	75.25	143 ± 29 ^a,b^	1.86 ± 0.81 ^b^	7.07 ± 3.43 ^b^	30.85 ± 4.72 ^b^
DPBAT:Cm II	2.5 ± 0.05 ^b^	64.59	133 ± 15 ^a^	0.98 ± 0.43 ^a^	2.38 ± 0.95 ^a^	39.95 ± 8.83 ^a^
DPBAT:Cm III	5.5 ± 0.05 ^d^	83.75	144 ± 23 ^a,b^	3.16 ± 0.38 ^a,b^	9.31 ± 1.05 ^a^	53.28 ± 10.38 ^a^

Mean—standard deviation of the analyses in triplicate. Values with the same letter in the same column do not show significant differences (*p* > 0.05) according to the Tukey test at a 95% confidence interval.

**Table 3 polymers-15-03106-t003:** Mass loss events (T1, T2 in the TGA/DTG) of the DPBAT, DPBAT:Cm I, DPBAT:Cm II, and DPBAT:Cm III films.

Films	Moisture (%)	*T_onset_* (°C)		Mass Loss (%)		Residue (%)
		1st	2nd	1st	2nd	
DPBAT	0.40	-	332.0	-	96.42	3.20
DPBAT:Cm I	1.89	267.3	347.6	16.92	77.73	3.46
DPBAT:Cm II	2.48	255.9	353.4	62.46	62.46	1.73
DPBAT:Cm III	2.49	262.1	340.4	24.73	68.65	4.13

## Data Availability

Data sharing is not applicable.

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
