# Peer review of "Effect of Addition of Cross-Linked Starch on the Properties of Degraded PBAT Poly(butylene adipate-co-terephthalate) Films"

_polymers, 2023, doi:10.3390/polym15143106_

Round 1
Reviewer 1 Report
1)Figure needs to be rechecked before acceptance. TGA and DSC figure specifically.
2)DSC and TGA table don't have standard deviation was it carried out once?
3)what about the color of films starch addition might have effect on its properties.
Author Response
1 - Figure needs to be rechecked before acceptance. TGA and DSC figure specifically.
Response: The reviewer is correct. The TG and DTG figures have been modified. DSC runs were not performed in this investigation.
2 - DSC and TGA table don't have standard deviation was it carried out once?
Response: TGA analysis was performed just once. Due to the very low variation of the intrinsic response of this technique, it does not make sense to perform three or more runs. DSC runs were not performed in this investigation.
3 - What about the color of films starch addition might have effect on its properties.
Response: As the colors of starch and PBAT are quite similar, the addition of cross-linked starch did not significantly affect the color of the blends obtained.
Reviewer 2 Report
In my opinion, the manuscript is well prepared from the organisational and content side. The conducted research is interesting and innovative both from the practical and scientific side. The authors have set up the research experiment correctly from the methodological side. The obtained results were thoroughly analysed and discussed. The results were statistically supported.
I accept the article in its present form.
Author Response
Responses to reviewer 2
1 - In my opinion, the manuscript is well prepared from the organisational and content side. The conducted research is interesting and innovative both from the practical and scientific side. The authors have set up the research experiment correctly from the methodological side. The obtained results were thoroughly analysed and discussed. The results were statistically supported.
I accept the article in its present form.
Response: The authors are grateful for the comments and the manuscript’s revision.
Reviewer 3 Report
Main problem of the subjected manuscript: in my opinion the title is a little bit misleading. Could you be so kind and reconsider the introduction of amendments.
I noticed, tha line numeration is not continous.
l. 100 only ref. 34 is valid.
Film elaboration. What was the moisture level of prepared mixes before extrusion. Also there is mentioned, that temperature dropped from 130 to 40C. So how was it performed? This sentence is a misleading, because, in first part it sis mentioned a pressure, and the temperature.
l.155 be more specific about "some algorithms" Please provide reference or change it.
2.3.4. Thickness and mechanical properties
This part will require some cosmetic changes. I suggest mention first about strips/belts preparation, then about thickness measurements
2.3.5 Please add reference
Results
In some places you add data related to PBAT. . Can you provide this informations for the all aspects as a part of results discussion.
Table 1 add "-" in Cm column.
Also notation PBATd: Cm starch 70:30 is better than DPBAT:Cm1. Or try to apply other abbreviation (Roman numbers??)
Conclusion
l. 118 films prepared with DPBAT ... I believe
and l. 119 degraded DPBAT - this time opposite situation (in DPBAT letter d stands for degraded)
no special remarks
Author Response
1 - Main problem of the subjected manuscript: in my opinion the title is a little bit misleading. Could you be so kind and reconsider the introduction of amendments.
Response: The title has been changed to suit the reviewer's suggestion.
2 - I noticed, the line numeration is not continous.
Response: Line numbering has been adjusted to suit the reviewer's suggestion.
3 - l. 100 only ref. 34 is valid.
Response: The reviewer is correct. Reference [34] is now pointed out as the pioneer reference of the starch modification method, and reference [33] is associated with the author who adapted the method for use in our laboratory.
4 - Film elaboration. What was the moisture level of prepared mixes before extrusion.
Response: This information was added in response to the reviewer's suggestion.
5 - Also there is mentioned, that temperature dropped from 130 to 40C. So how was it performed? This sentence is a misleading, because, in first part it sis mentioned a pressure, and the temperature.
Response: The pellets were subjected to a temperature of 130 °C for 15 minutes. After this period, the system temperature was set to 40 °C using the temperature controller. Upon reaching this temperature, the pressure between the presses was increased to 56 kPa. The text was modified to meet the reviewer's question.
6 - l.155 be more specific about "some algorithms" Please provide reference or change it.
Response: The software used to obtain the images and the contact angle value were added to the text to meet the reviewer's suggestion.
7 - 2.3.4. Thickness and mechanical properties
This part will require some cosmetic changes. I suggest mention first about strips/belts preparation, then about thickness measurements
Response: Modifications were made to the text describing the methodology of thickness measurements and mechanical properties to meet the reviewer's suggestion.
8 - 2.3.5 Please add reference
Response: The reference was added to meet the reviewer's suggestion.
9 - Results
In some places you add data related to PBAT. . Can you provide this informations for the all aspects as a part of results discussion.
Response: PBAT Information has been added to some parts of the results to address the reviewer's suggestion.
10 - Table 1 add "-" in Cm column.
Response: The Character "-" was added in Column Cm of Table 1 to meet the reviewer's suggestion.
11 - Also notation PBATd: Cm starch 70:30 is better than DPBAT:Cm1. Or try to apply other abbreviation (Roman numbers??).
Response: Roman numbers have been added to identify film formulations to meet the reviewer's suggestion.
12 - Conclusion
- 118 films prepared with DPBAT ... I believe
Response: The text was corrected as suggested by the reviewer.
13 - l. 119 degraded DPBAT - this time opposite situation (in DPBAT letter d stands for degraded)and
Response: The text was corrected as suggested by the reviewer.
Reviewer 4 Report
The manuscript is thoroughly prepared. The introduction leads well to the topic. Materials and methods are detailed and the experiment is possible to be reproduced. The discussion is properly lead, but could be prepared with more in depth focus. Nevertheless, the manuscript is lacking the reasoning why degraded PBAT was used both in the introduction as well as in discussion. The aothors should also analyze the degree of substitution of starch.
Detailed remarks:
Line 45-48 Source should be provided.
Line 48-49 corn should be also mentioned
Line 75 which exactly?
Line 76-80 What is the reason behind using degraded PBAT? The idea of using degraded polymer needs to be thoroughly justified.
Line 85 The polymer was stored exactly in 75% P.H.? If yes, state how (climate chamber, desiccator, etc.).
Line 87-88 were some these properties analysed prior and after storage?
Line 92-93 style
Line 104 homogenization ?
Line 110-111 the sentence is not clear
Line 180-181 as above
Figure 2 The images should be shown with the same magnification, currently there are misleading
Second part of the manuscript where the line mubering resets (page 7, after table 2)
Line 61 this is link to other research results, this should be indicated (the test conditions, investigated material may vary)
Line 103 – the degradation of starch is probably occurring only at T1 (regardless of the fact if the starch macromolecule was substituted with phosphate groups),
Line 118-119 This was not investigated, as only degraded films were analyzed.
The manuscript is readable, but some sentences are hard to read (style), as indicated in detailed comments.
Author Response
- The manuscript is thoroughly prepared. The introduction leads well to the topic. Materials and methods are detailed and the experiment is possible to be reproduced. The discussion is properly lead, but could be prepared with more in depth focus. Nevertheless, the manuscript is lacking the reasoning why degraded PBAT was used both in the introduction as well as in discussion. The authors should also analyze the degree of substitution of starch.
Response: The authors are grateful for the reviewer's considerations. During the most intense phase of the COVID-19 pandemic, researchers had to move away from research centers. As a consequence, many materials and inputs used for research have lost their quality. This happened with PBAT, which suffered the effects of degradation over time of storage. Thus, the need arose to develop alternatives to enable the reuse of this high added value polymer.
The degree of starch substitution has been added to the article.
- Detailed remarks:
- Line 45-48 Source should be provided.
Response: The sources used are included in the text with references [4 - 6]
- Line 48-49 corn should be also mentioned
Response: Corn was inserted to meet the reviewer's suggestion.
- Line 75 which exactly?
Response: PBAT degradation leads to loss of mechanical and thermal properties. This information was added to the article.
- Line 76-80 What is the reason behind using degraded PBAT? The idea of using degraded polymer needs to be thoroughly justified.
Response: The reason for using degraded PBAT was added to the introduction of the article.
- Line 85 The polymer was stored exactly in 75% P.H.? If yes, state how (climate chamber, desiccator, etc.).
Response: The polymer was stored in the manufacturer's own packaging. The sentence has been modified to meet the reviewer's suggestion.
Line 87-88 were some these properties analysed prior and after storage?
Response: The properties found for PBAT films degraded by long storage period and the properties reported in the literature of non-degraded PBAT films were compared.
Line 92-93 style
Response: The reviewer is correct. The sentence was modified.
Line 104 homogenization ?
Response: The sentence was modified to meet the reviewer's query.
Line 110-111 the sentence is not clear
Response: The sentence has been modified to make the text clearer
Line 180-181 as above.
Response: The sentence has been modified to make the text clearer
Figure 2 The images should be shown with the same magnification, currently there are misleading
Response: We know that the ideal is to use the same amplitude for all the images, however, different amplitudes were used so that it was possible to better visualize the interactions between cross-linked starch and PBAT.
Second part of the manuscript where the line nubering resets (page 7, after table 2)
Response: The page numbers and line numbering have been updated.
Line 61 this is link to other research results, this should be indicated (the test conditions, investigated material may vary)
Response: The information requested by the reviewer has been added to the text.
Line 103 – the degradation of starch is probably occurring only at T1 (regardless of the fact if the starch macromolecule was substituted with phosphate groups),
Response: The thermal decomposition T1 comprises glycerol and low molecular weight starch polymers such as amylose. Starch phosphorylation can increase the thermal stability of this biopolymer compared to its native counterparts when analyzed in native granular structures. The text was modified to improve readability.
Line 118-119 This was not investigated, as only degraded films were analyzed.
Response: The sentence was modified to meet the reviewer's suggestion.
Round 2
Reviewer 4 Report
The manuscript has been improved. The justification of degradation seems vague from scientific point of view. Nevertheless, the manuscript can be recommended for publciation in present form.